# Minispheroids as a Tool for Ligament Tissue Engineering: Do the Self-Assembly Techniques and Spheroid Dimensions Influence the Cruciate Ligamentocyte Phenotype?

**DOI:** 10.3390/ijms222011011

**Published:** 2021-10-12

**Authors:** Ingrid Zahn, Tobias Braun, Clemens Gögele, Gundula Schulze-Tanzil

**Affiliations:** 1Institute of Anatomy and Cell Biology, Paracelsus Medical University, Nuremberg and Salzburg, Prof. Ernst Nathan Str.1, 90419 Nuremberg, Germany; ingrid.zahn@fau.de (I.Z.); Tobias.Braun2@klinikum-nuernberg.de (T.B.); clemens.goegele@pmu.ac.at (C.G.); 2Institute of Functional and Clinical Anatomy, Friedrich Alexander University, Erlangen-Nuremberg, Universitätsstraße 19, 91054 Erlangen, Germany; 3Department of Applied Chemistry, Nuremberg Institute of Technology Georg Simon Ohm, Keßlerplatz 12, 90489 Nuremberg, Germany; 4Department of Cardiac Surgery (Cardiovascular Center), Klinikum Nürnberg Süd, Breslauer Str. 201, 90471 Nuremberg, Germany; 5Department of Biosciences, Paris Lodron University of Salzburg, Hellbrunnerstr 34, 5020 Salzburg, Austria

**Keywords:** cruciate ligament, spheroids, hanging drop, methyl cellulose

## Abstract

Spheroid culture might stabilize the ligamentocyte phenotype. Therefore, the phenotype of lapine cruciate ligamentocyte (L-CLs) minispheroids prepared either by hanging drop (HD) method or by using a novel spheroid plate (SP) and the option of methyl cellulose (MC) for tuning spheroid formation was tested. A total of 250 and 1000 L-CLs per spheroid were seeded as HDs or on an SP before performing cell viability assay, morphometry, gene expression (qRT-PCR) and protein immunolocalization after 7 (HD/SP) and 14 (SP) days. Stable and viable spheroids of both sizes could be produced with both methods, but more rapidly with SP. MC accelerated the formation of round spheroids (HD). Their circular areas decreased significantly during culturing. After 7 days, the diameters of HD-derived spheroids were significantly larger compared to those harvested from the SP, with a tendency of lower circularity suggesting an ellipsoid shape. Gene expression of decorin increased significantly after 7 days (HD, similar trend in SP), tenascin C tended to increase after 7 (HD/SP) and 14 days (SP), whereas collagen type 1 decreased (HD/SP) compared to the monolayer control. The cruciate ligament extracellular matrix components could be localized in all mini-spheroids, confirming their conserved expression profile and their suitability for ligament tissue engineering.

## 1. Introduction

Spheroids represent self-assembled three-dimensional (3D) cell aggregates with a round or ellipsoid shape. Spheroids allow long-term culture of primary cells, and hence, extracellular matrix (ECM) accumulation, intimate cell–cell as well as cell–ECM interactions. These cell–cell communications might stabilize the differentiated phenotype of primary cells. Spheroids can be produced using many cell types and techniques [1] including tenocytes and ligamentocytes [2,3,4]. They can be used as 3D tumor models for the screening of agents and strategies effective to inhibit tumorigenesis [1,5]. They represent a well-established method for stem cell multi-lineage differentiation [6]. Meanwhile, they are already used for biomaterial-free therapeutical strategies in the clinic, e.g., for cartilage repair (chondrospheres, spherox) [7,8]. In addition, they are applied for biomaterial-assisted tissue engineering to achieve directed cell seeding of scaffolds to develop novel approaches for regenerative medicine. Already many years ago, a spheroid-like in vitro tendon model based on high cell density was established which allowed tenocyte phenotype maintenance [9]. The phenotype stabilization of tenocytes cultured in this model was further confirmed by directly comparing it with monolayer (2D) and 3D scaffold culture models [10]. Tenocyte monolayer culture is known to be associated with tenocyte phenotypic changes and dedifferentiation [11,12,13]. Hoyer et al. investigated the expression profile of lapine anterior cruciate ligamentocytes in large spheroids consisting of 3 × 10^5^ cells, cultured under static and dynamic conditions in more detail and found distinct differences at the protein level induced by the respective culture conditions [14]. This suggests that the culture type of spheroids but possibly also the spheroid preparation technique and spheroid sizes might influence ligamentocytes expression profile. This might be of particular importance since spheroids provide promising approaches for tissue engineering. Spheroids, e.g., of adipose-tissue-derived stem cells, can be combined with electrospun scaffolds as building blocks for neotissue formation since spheroids fuse rapidly together [15,16]. Meanwhile, ligamentocyte spheroids were used in several previous studies as a valuable approach for scaffold seeding [14,17,18,19]. All these applications require high numbers of spheroids at a tunable size. For tissue engineering, the desired spheroid size depends on the pore size of the scaffolds. They should be small enough to enter scaffold pores, but large enough not to fall through the pores of the scaffold. Smaller spheroids may more easily adhere to the scaffold and cells might more rapidly emigrate from the spheroids and spread on the scaffold due to their larger relative surface to volume area compared to larger spheroids. In addition, the maximum spheroid size is limited by the requirement of sufficient nutrient diffusion into the core of the spheroid (the nutrition limit is around 250 µm) [20].

There exist many methods for spheroid production (Table 1: summarized for ligamentocytes, tenocytes, and MSCs) [1,21] with the hanging drop method being the longest and most widely used [16,18,19,22]. However, this technique is material-intensive, incubator-place, time-consuming, and not very suitable for long-term culturing.

The question arises whether another technique using a versatile spheroid plate with 750 conically shaped micro cavities within one macrowell might allow the high throughput production of 750 uniform cruciate ligamentocyte spheroids within one macrowell comparable to those acquired by the hanging drop method. Cruciate ligamentocyte viability, spheroid stability, size, and phenotype-related marker expression were compared dependent on the spheroid self-assembly method and spheroid sizes.

## 2. Results

### 2.1. Light Microscopical Morphology and Viability of Spheroids Produced by Hanging Drop and Spheroid Plate Techniques

Stable spheroids could be produced with 250 and 1000 cells by the hanging drop method (Figure 1A). After 48 h, round spheroids had assembled with both cell numbers. The spheroids formed by 250 and 1000 cells per spheroid were stable enough to be harvested after 48 h. Only a few dead cells could be detected in the spheroids produced with the hanging drop method (Figure 1B). There were no significant differences in the cell viability, comparing spheroids consisting of 250 and 1000 cells at each investigated time point. The ligamentocyte viability did not significantly differ during the entire investigation period and was never less than 97% (Figure 1C).

Using the spheroid plate, round spheroids became evident after 24 h (Figure 2A). They were sufficiently stable for harvesting after 24–48 h (Figure 2A). Spheroids reflected a high cell viability with only a few dead cells (Figure 2B). The comparison of spheroids consisting of 250 and 1000 cells harvested either at 7 or 14 days showed no significant difference in viability, independent of the cell number and time point of analysis (Figure 2C).

The viability, sizes, and circularity of the spheroids produced by both methods with both cell numbers were directly compared. Round and uniform spheroids were achieved with both techniques. There was no significant difference in the viability of spheroids produced with both methods, irrespective of the cell number used for spheroid assembly (Figure 3A,B).

The spheroid size was monitored since it is important for nutrition by diffusion. With both methods, a higher cell count led to a significantly larger spheroid size. Nevertheless, a significantly smaller size of the spheroids produced with the spheroid plate compared to the spheroids assembled in hanging drops became evident for both spheroid sizes (Figure 3C). The largest spheroids were produced with 1000 cells and the hanging drop method (157.0 ± 10.6 µm). There was a tendency for a decreased circularity in the spheroids of 250 and 1000 cells produced by the hanging drop method compared to spheroids derived from a spheroid plate (Figure 3D). This is also visible in Figure 1B. The hypothesized difference in shapes of spheroids produced by both methods in relation to the dimensions of the culture environments is summarized in Figure 3E.

Culturing spheroids in hanging drops or spheroid plates for a longer time (> 5–14 days) required the exchange of culture medium. This was difficult to perform in hanging drop culture and much easier to execute using the spheroid plate. Since many spheroids were required for RNA isolation (prepared on 15 petri dishes, each containing 30–50 HD per sample), the cultivation of HD-derived spheroids was limited to 7 days.

### 2.2. Size and Circular Surface Area of Spheroids Manufactured by the Hanging Drop or Spheroid Plate Method

Since it could be observed that spheroids became obviously smaller during culturing (Figure 1A and Figure 2A) the circular area depicted in the images taken from the spheroids was calculated at each day. There was a significant trend of decrease in circular areas, irrespective of the cell numbers used for spheroid formation and the technique applied (Figure 4A1–B2). In addition, using the spheroid plate, a significant decrease of circular area of spheroids (1000 cells per spheroid) could be detected comparing days 2 and 3, 4 and 5, and 6 and 7 (Figure 4A2). The process of compaction could be described for 1000 cells containing spheroids produced in the spheroid plate. The size difference between the calculated theoretical size and the real size of these spheroids was significant lesser at day 6 of the cultivation time (Appendix A).

### 2.3. Gene Expression of the Spheroids for Ligament-Related Components

Gene expression of typical ligament-related components such Collagen type 1 (COL1A1), decorin (DCN), tenascin C (TNC) could be demonstrated in spheroids produced by both methods (Figure 5). The expression profile was directly compared to the monolayer culture (P3-P6). There was generally no significant difference between the gene expression of spheroids produced by the hanging drop or spheroid plate methods at the investigated time point of 7 days, irrespective of whether 250 cells or 1000 cells containing spheroids were compared. However, in spheroids manufactured with 1000 cells by both techniques, there was a lower gene expression for COL1A1 compared to monolayer cultures (Figure 5). There was a significantly higher expression for DCN in spheroids (both sizes) produced by hanging drop compared to monolayers (Figure 5).

### 2.4. Localization of Ligament-Related ECM Components in the Spheroids

Typical ligament-related ECM components such as collagen type 1, decorin, and tenascin C were demonstrated in all spheroids by immunolabeling after 7 days (Figure 6). However, there were no major differences in regard to these components between spheroids manufactured by hanging drop or spheroid plate methods (Figure 6). Spheroids with a higher cell number (1000 cells) contained more unstained areas with only nuclear staining by DAPI. There were highly intense areas immunoreactive for collagen type 1. Tenascin C stained more centrally in the spheroids and decorin revealed a more peripherally localized immunoreactivity, which appeared to be more homogeneously distributed.

### 2.5. Effect of Methyl Cellulose on Spheroids Generated by Hanging Drop Culture

Methyl cellulose was added during spheroid formation as a possible tool to influence self-assembly of cells, and hence, spheroid formation. Experiments based on the HD method were performed with the addition of different concentrations of methyl cellulose. A low concentration of 0.05% methyl cellulose influenced formation of spheroids and their shapes (Figure 7A), whereas higher concentrations (0.1, 0.25, 0.5%) had an adverse effect (not shown). A significantly higher circularity of the spheroids until day 9 compared to the control was found using 0.05% methyl cellulose. The circularity in the controls increased at day 7 and 9 compared to the control at day 5 and 3 (Figure 7B). At day three, in the presence of methyl cellulose, the spheroid mean area was significantly lower compared to the control (Figure 7C). Alcian blue staining for visualization of cell distribution in inner parts of the spheroids and deposition of the ECM containing sulfated glycosaminoglycans displayed more intensive staining in the spheroids containing methyl cellulose. Some cell-free areas containing only ECM could be shown in controls and in the presence of methyl cellulose, but this was more pronounced in the control samples (Figure 7D).

## 3. Discussion

Spheroids are a versatile tool for tissue engineering and can be used as tunable building bricks for biofabrication of tissue models [16], reviewed by Laschke et al. [15]. In previous work, we could show that cruciate ligamentocyte spheroids are suitable for the colonization of synthetic polymer scaffolds [18,19,24]. When spheroids [4,22] or cell sheets [32] are seeded onto the scaffolds, cells emigrate and spread onto the scaffolds. However, high numbers of spheroids are required for tissue engineering; hence, high-throughput strategies for spheroid formation are needed. Spheroid adherence to a biomaterial and spreading of cells emigrating from larger spheroids on the biomaterial might be less effective than that observed in smaller ones. Hence, a strategy to produce larger amounts of smaller spheroids would be very helpful. The spheroid plate allows rapid production of 750 spheroids within one well of a 24-well plate. With the hanging drop method, around 15 dishes with 9 cm diameters are needed, each containing 50 spheroids to achieve the same number of spheroids. The conventional hanging drop method is widely used for spheroid production [2,4,14,16,19]. However, this culture has some further limitations. Using the hanging drop method, culture medium changes can barely be performed and are very time-consuming. For this reason, spheroids in the hanging drop culture could not be monitored as long as the spheroids in the spheroid plate. The spheroid plate method allows much easier cell culture medium changes. All 750 spheroids produced in one well of a spheroid plate are exposed to the same micro-milieu since they reside in the same culture chamber sharing their growth medium and cells can exchange soluble and extracellular vesicle-mediated paracrine mediators, which might support cell–cell communication. In contrast, the spheroid in each hanging drop has a completely isolated micro-milieu. The relation of the amount of culture media to cell number is in a hanging drop of 50 µL 18.75× higher compared to the cells in the spheroid plate under the condition of the experiments described here with 2 mL growth medium supernatant in each well. Nevertheless, the viability of cells remained high over time in spheroids produced by both methods with no significant differences.

Microscopical monitoring during spheroid formation and further culturing is much easier in the spheroid plate. However, there is a maximum size limitation of the spheroids produced by the spheroid plate method at 1000–2000 cells (dependent on the cell size of the respective cell type) per spheroid and on the dimension of the conic cavities of the plate as stated by the manufacturer and based on own experiments (Figure 3E). Resting on our own experience hanging drops with around 5 × 10^4^ cells containing spheroids can be performed [24].

The overall dimension of the spheroids in this study was small enough (< 250 µm diameter) to allow cell nutrition [20].

It is most important that the expression profile of ligamentocytes does not differ between spheroids produced by both methods. Since a non-adherent coating and another polymer (cycloolefin copolymer) compared to conventional culture dishes is used for the spheroid plate, it has to be determined whether this might affect the gene and protein expression of cells. However, in spheroids manufactured with 1000 cells by both techniques, there was a lower gene expression for COL1A1 detectable when compared to monolayer cultures. This was previously observed comparing another type of 3D culture, namely cell sheets, with monolayers using exactly the same cell type [32]. It could be explained by the fact that monolayer cells were deprived of their freshly produced ECM, which is rich in collagen type 1 during passaging, leading to a stimulation of the de novo collagen type 1 synthesis. In contrast, in spheroids, an accumulation and deposition of freshly produced collagen type 1 can be observed possibly inhibiting the collagen type 1 synthesis by negative feedback. The collagen type 1 deposition could be shown in all investigated types of spheroids by immunohistochemistry. In contrast, there was a significantly higher gene expression of DCN in spheroids (both sizes) produced by the hanging drop strategy compared to monolayers. The small proteoglycan is needed for collagen fibril and extracellular matrix organization [33]. Since a higher amount of the freshly produced collagen is refrained in the spheroids compared to the monolayers, the fibril assembly process might also be increased in the spheroids and induce decorin expression.

The observed decrease of spheroid sizes over culture time has also been reported for larger spheroids (3 × 10^5^ cells per spheroid) of the same cell type [14]. It describes the phenomena of spheroid compaction [34] mediated by more intense cell–cell contacts and probably emerging contractile forces of the cells applied to the ECM surrounding the cells. Compaction of spheroids with an increasing cell number could be observed here using both spheroid preparation methods. Therefore, the measured circular area of the spheroids manufactured with 1000 cells was compared with a hypothetically calculated fourfold volume expansion for the spheroids of 250 cells (Appendix A). Since this calculation is based on an ideal spheroid shape, the hanging drop method was excluded. The spheroids with 1000 cells from the spheroid plate showed a smaller circular area compared to the hypothetically calculated area of spheroids starting from 250 cells, which indicates a compaction of the spheroids with an increasing number of cells (Appendix A).

One risk during harvesting of spheroids is generally the fact that vital spheroids rapidly fuse when they come into direct contact. One advantage of the spheroid plate is that round spheroids form one day earlier compared to the hanging drop method. In the conic micro cavity, with the high slope of the wall, cells settle more rapidly down into the small deepest point of the cavity enforced to be in intensive cell–cell contact; hence, the cell–cell communication and compaction process occurs earlier. The micromilieu of the spheroids produced by hanging drops or in the spheroid plate substantially differs, which might influence the compaction process. The larger size of spheroids produced by the same cell number with the hanging drop compared to the spheroid plate method cannot easily be explained based on the investigations performed. It could be simply mocked by a more lense-like (hanging drop method) instead of a spherical shape (spherical plate method) (Figure 3E). The question of whether cell proliferation is higher in the spheroids produced in the hanging drop contributing to an increase in size compared to those harvested from the spherical plate remains to be addressed. Since stable spheroids assemble around 24 h later in the hanging drop, cells remain longer in suspension and might proliferate, whereas spheroids produced in the spherical plate start earlier with a maturation and differentiation process associated with a slowing down of cell proliferation. To modulate the spheroid formation in hanging drops, methyl cellulose was tested as a growth medium additive. Its effect on other cell types such as tumor cell lines has already been reported [35]. We could confirm for cruciate ligamentocytes that it indeed supports earlier spheroid formation and compaction in comparison to controls with no addition of methyl cellulose. The mode of action remains unclear. It might increase the viscosity of the growth medium and thereby enforce the cells to aggregate more rapidly. Obviously, it also captures sulfated glycosaminoglycans, which might be normally released in the growth medium, as shown by the more intense alcian blue staining.

## 4. Materials and Methods

### 4.1. Isolation of Lapine Cruciate Ligamentocytes

For isolation of lapine cruciate ligamentocytes anteroir (ACLs) and posterior cruciate ligaments (PCLs) of 10 healthy male and female New Zealand rabbits (between 4 and 24 months old) obtained from the abattoir were harvested. Slices of 1–2 mm of the ACLs and PCLs (Cls) were prepared and transferred into T25 culture flasks (Sarstedt AG & Co. KG, Nürnbrecht, Germany) with growth medium (Dulbecco’s Modified Eagle’s Medium (DMEM)/Ham’s F12 medium (1:1) (Bio&SELL, Feucht, Germany), containing 10% fetal bovine serum (FBS, Bio&SELL), 1% penicillin/streptomycin solution, 25 μg/mL ascorbic acid (Sigma-Aldrich, Munich, Germany), 2.5 μg/mL amphotericin B (Bio&SELL) and MEM amino acid solution (Sigma-Aldrich)).

Growth medium was changed 2–3 times a week. After 7–10 days, CL-derived ligamentocytes emigrated and were detached using 0.05% trypsin/ 0.02% ethylenediaminetetraacetic acid (EDTA) solution (Bio&SELL). The number of viable cells was determined using the trypan blue exclusion assay with a hemocytometer.

### 4.2. Spheroid Preparation

Spheroid self-assembly was mediated by the hanging drop method or using a micro cavity plate (SPHERICALPLATE 5D^®^, Kugelmeiers, Zollikerberg, Switzerland). For gene expression analysis, monolayers (2.4 × 10^3^ per cm^2^) were prepared as a control in a cell density normally used for cell expansion.

#### 4.2.1. Hanging Drops

For the hanging drop method, a cell suspension (250 or 1000 cells per 50 µL growth medium) was dropped on the inner surface of the lid of a non-coated 90 mm petri dish (nerbe plus GmbH & Co. KG, Winsen, Germany). The lid was turned back on the bottom of the dish filled with 15 mL phosphate-buffered saline (PBS, Bio&SELL) and cultured for up to 7 days (only in 4.2.2. for 9 days). To perform spheroid feeding with culture medium, (after 5 days) lids were turned again and 20 µL medium added per drop. Hanging drop cultures used for qRT-PCR experiments could only be maintained for 7 days since growth medium changes were risky and time consuming in around fifteen plates with 50 drops for each independent experiment.

#### 4.2.2. Hanging Drops with Methyl Cellulose

As an approach to possibly tune spheroid formation produced by the hanging drop method, 0.05% methyl cellulose (Würzteufel GmbH, Nagold, Germany) diluted in growth medium was added to the culture medium with the start of the hanging drop culture. Cultures without methyl cellulose served as controls. In this case, to allow an easier handling, larger spheroids (1 × 10^4^ cells per 50 µL drop) were prepared. Growth medium was changed after 5 days. Native cultures were monitored for 9 days and light microscopical images were taken from each drop at each time point of analysis using a Wild Heerbrugg Plan stereo microscope by a digital camera (Olympus OM-D, EM5-Mark II). For all independent experiments and every time point, 10 images of spheroids were taken. Images of spheroids of each day were analyzed morphometrically using ImageJ. A binary image was created using ImageJ; subsequently, the circularity and area were calculated.

#### 4.2.3. Spheroids Produced by Using the SPHERICALPLATE 5D^®^

The SPHERICALPLATE 5D^®^ is a 24-well plate. Two rows of wells (12 wells) on the plate (A1–A6 and C1–C6) contain 750 nanocoated cavities per well with a round-bottom shape allowing the production of 750 spheroids per well. Hence, per plate, 9000 microwells are available. The other 12 wells have a conventional flat surface suitable for monolayer culture. The plate is made of cycloolefin-copolymer. The wells were wetted with 1 mL PBS immediately before use, centrifuged at 300× *g* for 5 min to remove air bubbles, then PBS was removed and substituted by 0.5 mL culture medium before the cell suspension of the desired cell concentration was added (0.1875 × 10^6^ cells per well to achieve 250 cells per micro cavity/spheroid and 0.75 × 10^6^ cells per well to achieve 1000 cells per micro cavity/spheroid suspended in 1.5 mL culture medium). Light microscopical images were taken using an invert diavert microscope (Leitz, Wetzlar, Germany) and a digital camera (Olympus OM-D, EM5-MarkII).

### 4.3. Viability Assay

The cell viability in spheroids was visualized using a live/dead assay, which comprised the staining of the cells with fluorescein diacetate (FDA, Sigma-Aldrich) and propidium iodide (PI, Carl Roth GmbH and Ko.KG, Karlsruhe, Germany). Spheroids were incubated for 30 s in 50 µL FDA/PI staining solution (0.5% FDA and 0.1% PI dissolved in PBS). The green (living cells, FDA) or red (dead cells, PI) fluorescence was visualized using a SPEII confocal laser scanning microscope (CLSM, Leica Microsystems GmbH, Wetzlar, Germany).

The colonized area was measured based on the living cells with the ImageJ program. The cell viability was assessed based on the ratio of the area of the living cells compared to the area of the dead cells with the ImageJ program.

### 4.4. RNA Isolation from Ligamentocyte Spheroids Compared to the Monolayer

Spheroids (500 for hanging drop and 750 for spheroid plate culture) were snap-frozen after 7 days (each at least *n* = 3–6) and stored at −80 °C until homogenized in RLT-buffer containing 1% mercaptoethanol (Qiagen GmbH, Hilden, Germany). Monolayers (0.6 × 10^6^ ligamentocytes) cultured for 24 h in T25 flasks were incubated in RLT-buffer containing 1% mercaptoethanol and scraped off using a cell scraper (Sarstedt AG & Co. KG, Nürnbrecht, Germany). RNA was isolated using the RNeasy Mini kit according to the manufacturer’s instructions (Qiagen GmbH), with on-column DNA elimination. The purity and quantity of the RNA samples were analyzed using the Nanodrop ND-1000 spectrophotometer (Peqlab, Biotechnologie GmbH, Erlangen, Germany) applying the 260/280 absorbance ratio.

### 4.5. Quantitative Real Time PCR with cDNA Synthesized from Spheroid RNA

Total RNA was reversely transcribed into cDNA using the QuantiTect Reverse Transcription Kit (Qiagen GmbH) according to the manufacturer’s protocol. Equal amounts of cDNA were applied for each quantitative real-time PCR (qRT-PCR) reaction performed with the TaqMan Gene Expression Assays (Life Technologies: Applied Bioscience (ABI), Foster City, CA, USA). Primer pairs for collagen type 1 (COL1A1), decorin (DCN), tenascin (TNC), and the reference gene glycerin-aldehyde-3-phosphate-dehydrogenase (GAPDH) were used (Table 2). qRT-PCR was performed using the real-time PCR detector StepOnePlus (ABI) thermocycler with StepOnePlus software 2.3 (ABI). The relative expression of the genes of interest by cells in spheroids was normalized to the GAPDH expression and calculated as recommended [36].

### 4.6. Immunolabeling of Ligament-Associated Components in Spheroid Culture

The protein expression of ligament-related components was assessed using immunolabeling and CLSM. Spheroids were fixed for 15 min in 4% paraformaldehyde (PFA), washed in phosphate-buffered saline and incubated in blocking buffer (5% diluted in TBS (TBS: 0.05 M TRIS, 0.015 M NaCl, pH 7.6), with 0.1% Triton X 100 for cell permeabilization) for 20 min at room temperature (RT). Then, spheroids were immersed in primary antibody suspension (collagen type 1 (1:30, goat-anti-human), decorin (1:50, rabbit-anti-human) and tenascin C (1:50, mouse-anti-human) (Table 3) overnight at 4 °C in a humid chamber. As negative controls, the primary antibodies were omitted. Spheroids were rinsed with PBS before incubation with donkey-anti-mouse, -goat cyanine (Cy) 3 coupled or anti-rabbit Alexa-Fluor488 labeled secondary antibodies (diluted 1:200 in TBS) overnight at 4 °C in a humid chamber. Cell nuclei were counterstained with 10 µg/mL 4′,6′-diamidino-2-phenylindol (DAPI, Roche, Mannheim, Germany). Labeled spheroids were washed three-times for 5 min with PBS before being directly examined using the CLSM.

### 4.7. Alcian Blue Staining of Spheroids Produced by the Hanging Drop Culture

The spheroids were fixed in 4% paraformaldehyde (PFA) for 15 min and immersed in HistoGel (ThermoFisher Scientific Inc., Darmstadt, Germany) before being paraffin embedding. Sections of 7 µm were deparaffinized in xylol for 10 min (Carl Roth GmbH and Ko.KG) and rehydrated with a descending ethanol row (ETOH, 99.8%, 96%, 80%, 70%) (Carl Roth GmbH and Ko.KG).

For the alcian blue (AB) stain, spheroid sections were incubated for 3 min in 1% acetic acid before being stained for 30 min in 1% AB staining solution (Carl Roth GmbH and Ko.KG). After rinsing in 3% acetic acid and followed by a washing step (2 min) in distilled water, cell nuclei were counterstained with nuclear fast red aluminium sulphate solution (Carl Roth GmbH and Ko.KG) for 5 min. AB-stained sections were embedded in Entellan (Merck KGaA, Darmstadt, Germany). Images were taken using a light microscope (DM1000 LED, Leica Microsystems GmbH).

### 4.8. Determination of Spheroid Diameter and Circularity

The spheroid diameter and circularity were determined using the images of the live/dead assay. The diameter of the spheroids was measured based on a vertical line through the centre of each spheroid. At least 35 spheroids were measured for each data point. ImageJ was used to determine the circularity of the individual spheroids.

### 4.9. Statistical Analysis 

Data were expressed as mean values with standard deviations. Statistics were performed using Graphpad Prism 8 (version 8.43, GraphPad Software, San Diego, CA, USA). Statistical significance was set at a *p* value of ≤ 0.05. Normalized (based on the monolayer: gene expression analysis) and unnormalized data was evaluated using one-way ANOVA with multiple comparison (Figure 7). The Grubbs test was applied to exclude outliers. A test for linear trends was performed to measure a systematic increase (or decrease) between different time points.

## 5. Conclusions

Both methods applied are suitable to produce very small and stable mini-spheroids from ligamentocytes depicting a ligament-related expression profile with no significant difference depending on the assembly technique. The described spheroid plate provides an easy-to-perform high-throughput method suitable for ligamentocyte spheroid assembly, as shown in the present study. It seems to be a sophisticated tool to harvest a high number of reproducible spheroids well tunable in size for tissue engineering approaches. Supplements such as methyl cellulose can be used to optimize the L-CL spheroid formation.

## Figures and Tables

**Figure 1 ijms-22-11011-f001:**
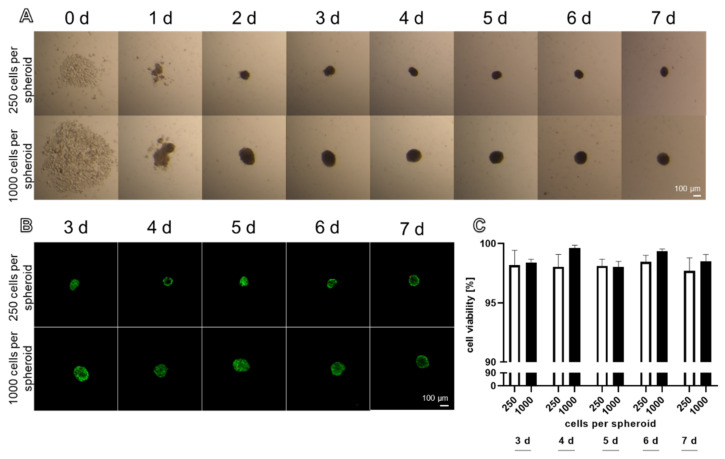
Ligamentocyte spheroid formation and viability based on 250 and 1000 cells per spheroid documented over 7 days using the hanging drop method. (**A**) Invert microscopical images of the same spheroids over 7 days. (**B**) Viability of cells cultured in spheroids prepared by the hanging drop method. Dead cells are red stained with propidium iodide and living cells are green due to metabolizing fluorescein diacetate. Scale bars: 100 µm (**A**,**B**). (**C**) Viability was monitored by CLSM after 3 to 7 days and calculated using ImageJ. Statistics: *n* = 3 independent experiments were performed with cells of five different donors. One-way ANOVA (Tukey’s multiple comparisons test) for comparison between groups (**C**).

**Figure 2 ijms-22-11011-f002:**
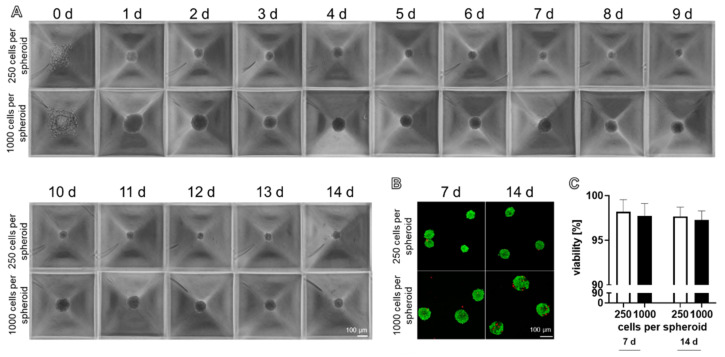
Ligamentocyte spheroid formation and viability based on 250 and 1000 cells per spheroid documented over 14 days using the spheroid plate. (**A**) Invert microscopical images of the spheroids within the micro cavities of the plates. (**B**) Viability of cells cultured in the spheroid plate at 7 and 14 days. Dead cells stained with propidium iodide are red and living cells metabolizing fluorescein diacetate are green. Scale bars: 100 µm. (**C**) Viability was monitored by CLSM and calculated using ImageJ. Statistics: *n* = 5 independent experiments were performed with cells of five different donors. One-way ANOVA (Tukey’s multiple comparisons test) for comparison between groups (**C**).

**Figure 3 ijms-22-11011-f003:**
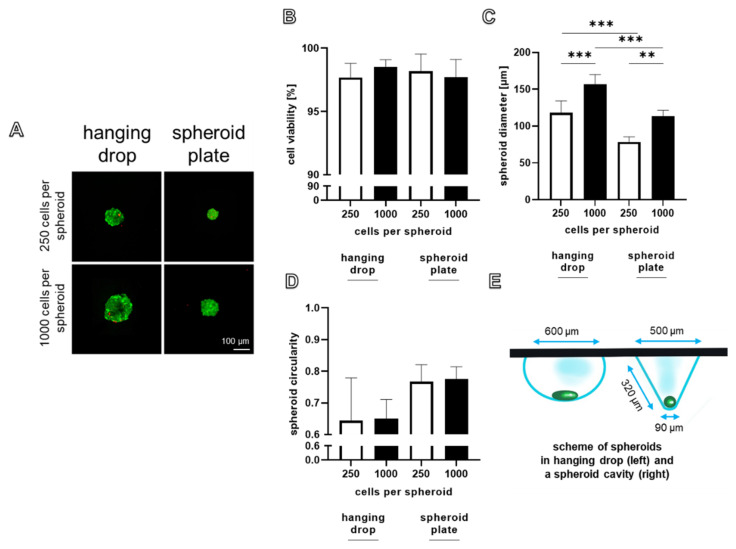
Direct comparison of viability and sizes of spheroids produced with the hanging drop and spheroid plate methods using either 250 or 1000 cells per spheroid at day 7. (**A**) Dead cells stained with propidium iodide are red and living cells metabolizing fluorescein diacetate are green. Viability was monitored by CLSM after 7 days. Scale bar: 100 µm. (**B**) The number of dead and viable cells was calculated using ImageJ. (**C**) Spheroid diameters were measured using the CLSM. (**D**) Spheroid circularity was defined using ImageJ. (**E**) Schematic representation of spheroids in a hanging drop (50 µL) and a spheroid cavity. Statistics: *n* = 3-5 independent experiments were performed with cells of three different donors. One-way ANOVA (Tukey’s multiple comparisons test) for comparison between groups. *p* values: ** < 0.01, *** < 0.001 (**B**,**C**,**D**).

**Figure 4 ijms-22-11011-f004:**
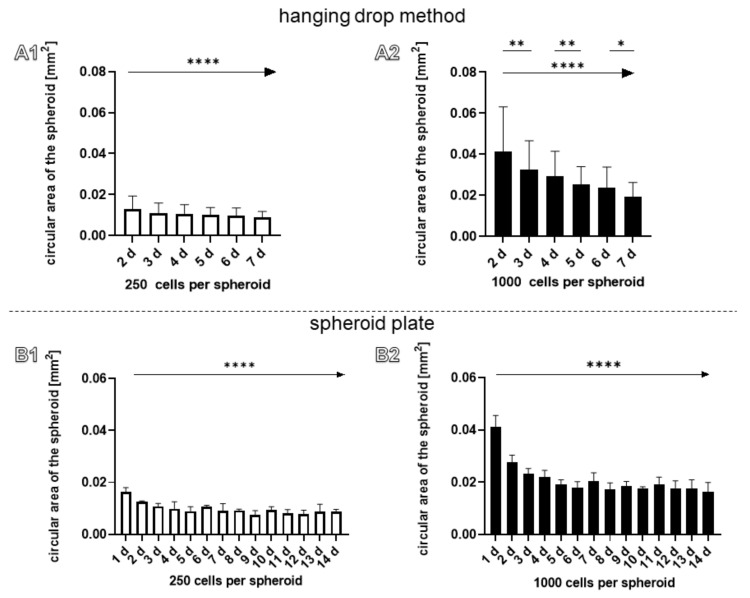
The circular surface area was measured in spheroids prepared either by the hanging-drop- (**A1**,**A2**) or spheroid-plate-based methods (**B1**,**B2**). (**A1**,**B1**) 250 cells per spheroid. (**A2**,**B2**) 1000 cells per spheroid. Statistics: *n* = 3 independent experiments were performed with cells of three different donors. One-way ANOVA (Tukey’s multiple comparisons test) for comparison between groups. Linear trend test (*→ *p* values: * <0.05, ** <0.01, **** <0.0001 (**A1**–**B2**).

**Figure 5 ijms-22-11011-f005:**
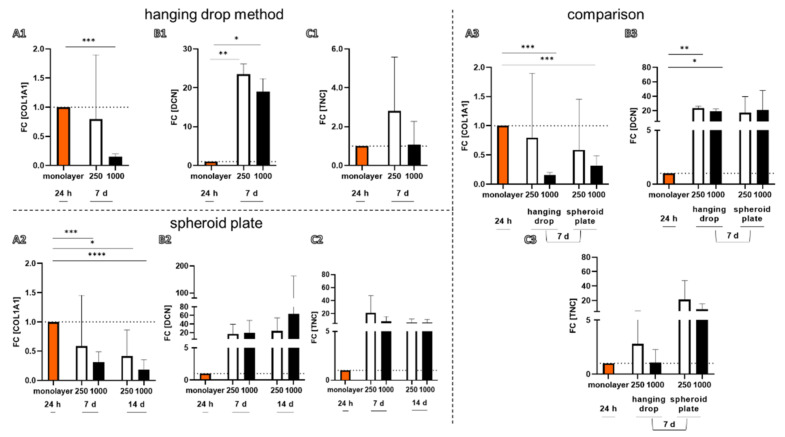
Gene expression in spheroids prepared in hanging drops or using the spheroid plate in comparison to that in monolayer culture. (**A**) Collagen type 1 (COL1A1), (**B**) decorin (DCN), (**C**) tenascin C (TNC). (**A1**–C**1**) Gene expression of ligamentocyte spheroids prepared by the hanging drop method after 7 days. (**A2**–**C2**) Gene expression after 7 and 14 days in spheroids prepared with the spheroid plate. (**A3**–**C3**) Direct comparison of the gene expression in spheroids prepared by both methods after 7 days. Statistics: *n* = 5 independent experiments were performed with cells of five different donors. One-way ANOVA (Tukey’s multiple comparisons test) for comparison between groups. Grubbs test (α = 0.05). *p* values: * < 0.05, ** < 0.01, *** < 0.001, **** < 0.0001 (**A1**–**C3**).

**Figure 6 ijms-22-11011-f006:**
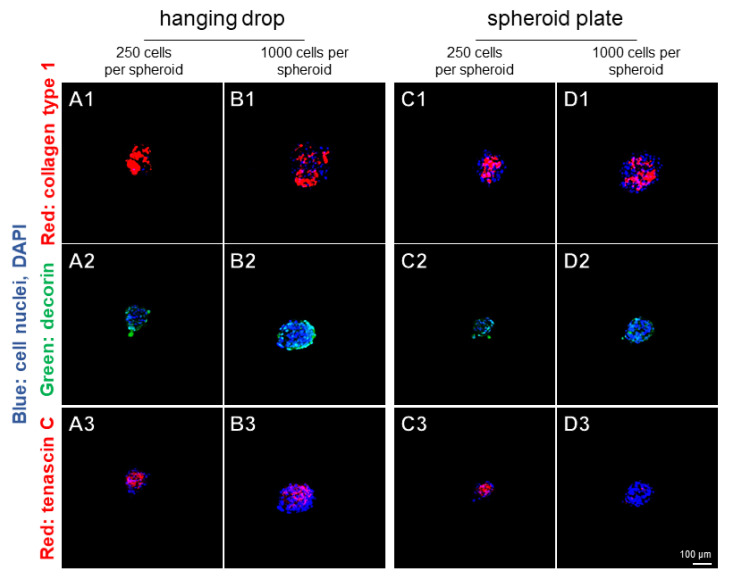
Distribution of ligament-related ECM components in spheroids prepared in hanging drops or using the spheroid plate after 7 days. (**A1**–**D1**) Collagen type 1 (red), (**A2**–**D2**) decorin (green) and (**A3**–**D3**) tenascin C (red). Cell nuclei were counterstained using 4’,6-diamidino-2-phenylindole (DAPI). Hanging-drop-derived spheroids consisting of 250 (**A1**–**A3**) or 1000 (**B1**–**B3**) ligamentocytes. Spheroids prepared by the spheroid plate consisting of 250 (**C1**–**C3**) or 1000 (**D1**–**D3**) ligamentocytes. Representative images are shown from three independent experiments. Scale bar: 100 µm.

**Figure 7 ijms-22-11011-f007:**
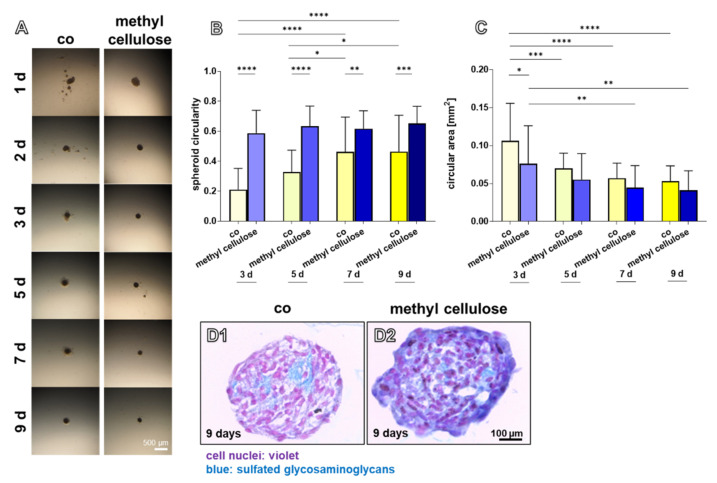
Effect of methyl cellulose on spheroids produced with the HD methods and monitored for 9 days. (**A**) native spheroids are shown after different culturing days in hanging drops (1–3, 5, 7, and 9 days). (**B**) spheroid circularity, (**C**) circular area (co: control, yellow bars). (**D1–2**) Alcian blue staining of a central section of the control (**D1**) spheroids and those (**D2**) treated with methyl cellulose. Spheroids consisted of 1 × 10^4^ cells. 10 spheroids per day and per experiment were analyzed. Representative images are shown derived from three independent experiments. Scale bars: 500 µm (**A**), 100 µm (**D1**,**D2**). Statistics: *n* = 3 independent experiments were performed with cells of three different donors. One-way ANOVA (Tukey’s multiple comparisons test) for comparison between groups. *p* values: * <0.05, ** <0.01, *** <0.001, **** < 0.0001 (**B**,**C**).

**Table 1 ijms-22-11011-t001:** Synopsis of techniques used for spheroid formation based on ligamentocytes, tenocytes, and stem cells.

Method	Spheroid Formation Technique	Cell Type	Advantage	Disadvantage	Reference
Hanging Drop	no surface to adhere	Ligamentocyte, tenocyte, ADSC	-no special device required -cost effective -uniform spheroids -of very small (e.g., 250 cells) and large sizes (50.000 cells)	-work intensive -not suitable for all cell types -no long-term culture (medium changes challenging) -size limit (drop size)	[16,18,22,23]
Pellets	centrifugal forces aggregate cells	SV40 modified ligamentocyte	-rapid -cost effective -reproducible -larger sizes possible	-shear forces -production of higher numbers difficult -nutrition of larger spheroids limited	[24]
Agar overlay-technique (hydrogels) or low attachment 96 well plates	non adherent surfaces, e.g., using hydrogels	Ligamentocyte, osteoblast, fibroblast endothelial cell co-culture	-long-term culture possible -larger sizes possible	-spheroid harvest after culturing difficult -no production of higher spheroid numbers	[14,25,26]
Roller bottle, spinner flask	continuous rotation prevents cell adherence	MSC	-larger size possible -not so expensive -long-term culture possible -dynamic culture	-size differences -shear forces -large volume of growth medium	[27]
Random positioning machine	prevention of gravidity and hence, cell adherence	Tenocyte	-larger spheroids -long-term culture possible -dynamic culture	-expensive device -no production of higher spheroid numbers	[3,28]
Micro cavity plates	-non adherent surface and conic shape (Aggrewell ^TM^ inserts and SPERICAL PLATE 5D^®^)	Ligamentocyte, stem cells	-high throughput -spheroid size can be selected within a wide range (100–2000) -reproducible -long-term culture possible	-expensive -spheroid size limit (micro cavity dimension)	[6,24,29]
Magnetic levitation	Magnetic particles associated with cells are aggregated by applying a magnetic field	MSC	-rapid spheroid formation -long-term culture	-magnetic field and particles might affect cells -cells must be able to phagocytize the magnetic nanoparticles	[30,31]

ADSC: adipose-tissue-derived stem cell, MSC: mesenchymal stem cell, SV40: simian vacuolating virus 40.

**Table 2 ijms-22-11011-t002:** Primers used to assess gene expression.

Gene Symbol	Species	Gene Name	NCBI Gene Reference	Efficacy	Amplicon Length (bp)	Assay ID *
COL1A1	*Oryctolagus cuniculus*	collagen type 1	AY633663.1	1.94	70	Oc03396073_g1
DCN	*Homo sapiens*	decorin	NM_133503.3	2.03	77	Hs00370384_m1
GAPDH	*Oryctolagus cuniculus*	glycerinaldehyde-3-phosphate-dehydrogenase	NM_001082253.1	1.95	82	Oc03823402_g1
TNC	*Oryctolagus cuniculus*	tenascin C	FJ480400.1	1.83	61	Oc06726696_m1

* All primers from Applied Biosystems^®^ (Life technologies ^TM^).

**Table 3 ijms-22-11011-t003:** Antibodies used to localize proteins.

Target	Primary Antibody	Dilution	Secondary Antibody	Dilution
collagen type 1	goat-anti-human COL1A1, Abcam, Cambridge, UK	1:30	donkey-anti-goat; cy3, Dianova GmbH, Hamburg, Germany	1:200
decorin	rabbit-anti-human, OriGene EU, DE	1:50	donkey-anti-rabbit; Alexa-Fluor488, ThermoFisher Scientific Inc., Germany	1:200
tenascin C	mouse-anti-human LSBio, Seattle, WA, USA	1:50	donkey-anti-mouse; cy3, Dianova GmbH, Hamburg, Germany	1:200

## Data Availability

Data supporting the reported results can be provided by contacting the first author.

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
