# Peer review of "Minispheroids as a Tool for Ligament Tissue Engineering: Do the Self-Assembly Techniques and Spheroid Dimensions Influence the Cruciate Ligamentocyte Phenotype?"

_ijms, 2021, doi:10.3390/ijms222011011_

Round 1

Reviewer 1 Report

Review for IJMS

Minisperoids as a tool for ligament tissue engineering:

Do the self-assembly techniques and spheroid dimensions influence cruciate ligamentocyte phenotype?

The authors provide a good comparison of hanging drop-produced minispheroids of either 250 or 1000 ligamentocytes with the production of such microbodies in spherical plates. I think it is technical paper that is highly practical because one can decide upon which technique might suit the personal objectives best.

I have some questions and some minor issues.

  1. Why were the minispheroids cultured for so long time periods, such as 7 or even 14 days, when they are already formed after 48 hours?

  1. Why were ligamentocytes of P3-6 (line 165) used, despite the fact of well-known phenotypic drift for >P3?

  1. The limited size of 250 µm diameter for the nutrition was mentioned several times in the manuscript, for example on line 244; where does this limit come from? There was no reference. Was it determined for this paper?

  1. Line 210; “…cells emigrate and spread onto the scaffolds”. Please cite Schneider, I., et al. (2019). "3D microtissue–derived human stem cells seeded on electrospun nanocomposites under shear stress: Modulation of gene expression." Journal of the Mechanical Behavior of Biomedical Materials 102: 103481.

  1. Line 261; it was not clear to me why DNC was more expressed in spheroids than in normal cell culture of ligamentocytes – the explanation given was not clear enough. Please re-phrase to make it clearer.

  1. All figures were well designed and the results were clear. For Figure 5, typical marker genes such as Mohawk or Scleraxis are missing. If the authors have cDNA left over from these experiments, it would be highly interesting to verify the expression and to compare the cultivation techniques with respect to these markers as well.

Author Response

Reviewer 1:

The authors provide a good comparison of hanging drop-produced minispheroids of either 250 or 1000 ligamentocytes with the production of such microbodies in spherical plates. I think it is technical paper that is highly practical because one can decide upon which technique might suit the personal objectives best.

Response: we thank the reviewer for the encouraging comment.

  1. Why were the minispheroids cultured for so long time periods, such as 7 or even 14 days, when they are already formed after 48 hours?

Response: we wanted not only to produce stable spheroid but also to test the influence of the environment: either the drop of culture medium in the hanging drop culture or the contact to a special very translucent polymer of which the spheroid plates are made for microscopy reasons. Since a 3D spheroid model is described, researcher might use it in future also for tenogenic/ ligamentogenic differentiation which requires a longer 3D culturing period. The long period should show whether the cells might transdifferentiate into an aberrant differentiation lineage. 48 h reflect only the self aggregation phase. The days thereafter indicate the phase of ECM deposition and specific differentiation.

  1. Why were ligamentocytes of P3-6 (line 165) used, despite the fact of well-known phenotypic drift for >P3?

Response: The ligamentocytes were usually at P1 cryopreserved and have to be sufficiently expanded to achieve the large number of cells required for parallel 3D experiments with both models e.g. for gene expression analyses. Hence, we had to face the compromise to include also higher passage cells.

  1. The limited size of 250 µm diameter for the nutrition was mentioned several times in the manuscript, for example in line 244; where does this limit come from? There was no reference. Was it determined for this paper?

Response: The nutrition limit by diffusion into spheroids was indeed extracted from literature and not based on own experience. The reference was cited on page 3. Griffith LG, Swartz MA. Capturing complex 3D tissue physiology in vitro. Nat Rev Mol Cell Biol. 2006;7(3):211-24.

We added it now also on pages 13, line 245.

  1. Line 210; “…cells emigrate and spread onto the scaffolds”. Please cite Schneider, I., et al. (2019). "3D microtissue–derived human stem cells seeded on electrospun nanocomposites under shear stress: Modulation of gene expression." Journal of the Mechanical Behavior of Biomedical Materials 102: 103481.

 Response: we thank the reviewer for the good advice and cite the recommended reference now.

  1. Line 261; it was not clear to me why DNC was more expressed in spheroids than in normal cell culture of ligamentocytes – the explanation given was not clear enough. Please re-phrase to make it clearer.

 Response: we provide a clearer hypothesis for the decorin expression upregulated in spheroids now.

  1. All figures were well designed and the results were clear. For Figure 5, typical marker genes such as Mohawk or Scleraxis are missing. If the authors have cDNA left over from these experiments, it would be highly interesting to verify the expression and to compare the cultivation techniques with respect to these markers as well.

Response: For the analyses at the gene expression level we had limitations to get enough RNA from the hanging drop samples. We could only perform Mohawk gene expression analysis for the spheroids consisting of 1000 cells, but could not compare it with the HD spheroids. Hence, we could not include this data. It was still detectable after 14 d in the spheroids, did not differ significantly between spheroids and monolayers of the same time point of analysis but was significantly lower compared with the culture initially used for spheroid preparation.

Figure 1: Gene expression of Mohawk (MKX) in spheroids prepared with the spheroid plate after 7 and 14 days.

Reviewer 2 Report

The manuscript “Minispheroids as a tool for ligament tissue engineering: Do the self-assembly techniques and spheroid dimensions influence cruciate ligamentocyte phenotype?” authored by Zahn et al., represents an interesting work that addresses the effect of spheroid-formation technique on the ligamentocyte phenotype.

The manuscript is well written and the experimental design is convincing. However, the Authors should address some critical points in order to consider the manuscript suitable for publication.

Critical points:

-Paragraph 1.4:  The authors tested different concentrations of methylcellulose as reported in lines 191-193. Please, it is suggested to specify exactly the different methylcellulose concentrations used.

-Figure 7: The spheroid formed in the methylcellulose at day 3 shows low circularity compared to respective spheroid in co (Fig 7A). However, in the histogram Fig.7B at day 3, the values reported do not seem to reflect the images. In addition, values for day 1 and 2 are not reported in the Fig.7B and C. Please, the improvement of Figure 7 is required.

Minor points:

-Paragraph 1.3. “Localization of ligament-related ECM components in the spheroids.” The used terms Collagen type I, Tenascin C, etc… should be harmonized throughout the text. Please adjust.

- Some typo-errors are present in the text. Please, correct.

Author Response

Reviewer 2

Open Review

The manuscript “Minispheroids as a tool for ligament tissue engineering: Do the self-assembly techniques and spheroid dimensions influence cruciate ligamentocyte phenotype?” authored by Zahn et al., represents an interesting work that addresses the effect of spheroid-formation technique on the ligamentocyte phenotype.

The manuscript is well written and the experimental design is convincing. However, the Authors should address some critical points in order to consider the manuscript suitable for publication.

 Response: we thank the reviewer for the encouraging comment.

Critical points:

-Paragraph 1.4:  The authors tested different concentrations of methylcellulose as reported in lines 191-193. Please, it is suggested to specify exactly the different methylcellulose concentrations used.

 Response: we provided the concentrations tested now, before selecting 0.05 for the study. It has been mentioned now in section 1.4.

Figure 2: Influence of methyl cellulose on spheroid formation. The images show the spheroid formation by ligamentocytes after 3 days in culture. In contrast to 0.05% methyl cellulose (MC) in the presence of higher MC concentrations (0.1-0.5%) ligamentocytes did not accomplish spheroid formation. Interestingly, the height of the concentration correlated with the number of „satellite“ spheroids. Only 0.05% MC resulted in formation of round and very compact spheroids. (0.5% and 0.25% pilot experiment: n=1, 0.1% and 0.05% in n=3).

-Figure 7: The spheroid formed in the methylcellulose at day 3 shows low circularity compared to respective spheroid in co (Fig 7A). However, in the histogram Fig.7B at day 3, the values reported do not seem to reflect the images. In addition, values for day 1 and 2 are not reported in the Fig.7B and C. Please, the improvement of Figure 7 is required.

Response: We apologize for this mistake. Both figure rows were indeed unfortunately transposed. We corrected it now.

Minor points:

-Paragraph 1.3. “Localization of ligament-related ECM components in the spheroids.” The used terms Collagen type I, Tenascin C, etc… should be harmonized throughout the text. Please adjust.

Response: Done.

- Some typo-errors are present in the text. Please, correct.

Response: Done.

Round 2

Reviewer 2 Report

The Authors  responded by satisfying the Reviewer's requests.
The Manuscript is interesting and now well structured.

The Manuscript is now suitable for publication.